# A Novel Carrier Loop Based on Coarse-to-Fine Weighted Adaptive Kalman Filter for Weak Communication-Positioning Integrated Signal

**DOI:** 10.3390/s22114068

**Published:** 2022-05-27

**Authors:** Xiwen Deng, Zhongliang Deng, Jingrong Liu, Zhichao Zhang

**Affiliations:** School of Electronic Engineering, Beijing University of Posts and Telecommunications, No. 10 Xitucheng Road, Haidian District, Beijing 100876, China; dengzhl@bupt.edu.cn (Z.D.); jingrongliu@bupt.edu.cn (J.L.); 2020110459@bupt.cn (Z.Z.)

**Keywords:** communication-positioning integrated signal, carrier tracking, adaptive filtering

## Abstract

We propose a communication-navigation integrated signal (CPIS), which is superimposed on the communication signal with power that does not affect the communication service, and realizes high-precision indoor positioning in a mobile communication network. Due to the occlusion of indoor obstacles and the power limitation of the positioning signal, existing carrier loop algorithms have large tracking errors in weak signal environments, which limits the positioning performance of the receiver in a complex environment. The carrier loop based on Kalman filtering (KF) has a good performance in respect of weak signals. However, the carrier frequency error of acquisition under weak signals is large, and the KF loop cannot converge quickly. Moreover, the KF algorithm based on fixed noise covariance increases or diverges in filtering error in complex environments. In this paper, a coarse-to-fine weighted adaptive Kalman filter (WAKF)-based carrier loop algorithm is proposed to solve the above problems of the receiver. In the coarse tracking stage, acquisition error reduction and bit synchronization are realized, and then a carrier loop based on Sage–Husa adaptive filtering is entered. Considering the shortcomings of the filter divergence caused by the negative covariance matrix of Sage–Husa in the filter update process, the weighted factor is given and UD decomposition is introduced to suppress the filtering divergence and improve the filtering accuracy. The simulation and actual environment test results show that the tracking sensitivity of the proposed algorithm is better than that based on the Sage–Husa adaptive filtering algorithm. In addition, compared with the weighted Sage–Husa AKF algorithm, the coarse-to-fine WAKF-based carrier loop algorithm converges faster.

## 1. Introduction

With the introduction of concepts such as 5G/6G, the Internet of Things, smart cities, and autonomous driving, high-precision location information has become a key factor driving their development. The Global Navigation Satellite System (GNSS) has provided high-precision outdoor location services, but it is limited by the occlusion of buildings and cannot be positioned indoors [1,2,3]. Positioning technology based on Bluetooth, Wi-Fi, and ultra-wide band (UWB) can provide good indoor location services, but the cost of large-scale coverage is high [4,5,6,7,8]. The dense infrastructure of mobile communication base stations provides the possibility for wide-area indoor and outdoor high-precision positioning. Positioning techniques based on 2G–5G cellular networks such as signal strength, signal propagation time, and angle of arrival have been extensively studied [9,10,11]. Fingerprint matching is used in mobile communication base station positioning, but due to the complex environment between stations, the positioning error ranges from ten meters to tens of meters [9]. 5G uses millimeter waves with high directivity and large-scale antennas, which can achieve better angle-of-arrival (AOA) positioning accuracy, but the positioning accuracy is limited by the size of the antenna [12,13]. The positioning reference signal (PRS) is a signal designed for positioning in the 4G network. It needs to occupy the time-frequency resources of communication services, which will cause the communication quality to deteriorate. At the same time, due to the discontinuous broadcast of PRS, the positioning accuracy is only tens of meters [14].

We propose a communication-positioning integrated system based on mobile communication networks to achieve high-precision positioning based on time difference of arrival (TDOA) [15,16]. The communication-positioning integrated signal (CPIS) is based on the non-orthogonal multiple access (NOMA) principle, in which the positioning signal is superimposed on the communication signal and broadcast in the form of low power. CPIS meets the following requirements: (1) the positioning signal does not interfere with the communication, i.e., the time-frequency resources of positioning signal multiplexing communication must be based on the premise of ensuring the quality of service (QoS) of communication, and do not occupy time-frequency resources alone; (2) the power of positioning signals for different users can be configured, i.e., interference between users is reduced by configuring different positioning signal powers; (3) the positioning signal is continuous, i.e., the positioning signal adopts direct sequence spread spectrum-code division multiple access (DSSS-CDMA), and the positioning receiver tracks continuous signals to achieve high-precision ranging. Interference cancellation technology is used to solve the interference of the strong communication signal to the weak positioning signal, and the positioning signal can be well recovered by reconstructing the weak signal. This technique of superimposing positioning and communication signals has been used in the China Mobile Multimedia Broadcasting (CMMB) system and has achieved fine wide-area indoor and outdoor positioning [17,18,19,20]. Considering the dense networking and strong signal coverage of communication base stations, we extend the technology to mobile communication networks.

Due to the complex indoor environment, the signal is easily attenuated by the environment, and at the same time, in order not to interfere with the communication, the transmission power of the positioning signal is limited. The weak signal affects the performance of the receiver to track the positioning signal. Therefore, we need to design a high-sensitivity positioning receiver. This paper focuses on the carrier tracking loop that is extremely fragile under weak signal conditions, and its threshold limits the unaided receiver performance [21]. Similar to GNSS receivers, CPIS receivers implement carrier tracking through frequency-locked loop (FLL) or phase-locked loop (PLL) so as to realize carrier synchronization [22,23]. Increasing the sensitivity by increasing the coherent integration time means that the loop noise bandwidth is set relatively small, which limits the dynamic adaptability of the loop. Traditional FLLs and PLLs have difficulty maintaining stable locks in harsh environments [24]. A tracking loop based on a Kalman filter (KF) can dynamically adjust the loop bandwidth and improve the tracking sensitivity [25]. However, the KF relies on the prior statistics of noise covariance, and it is difficult to obtain accurate statistics in practical applications. Since the noise variance remains constant during the filtering process, the KF tracking loop will diverge due to the filtering of environmental changes. Real-time estimation of noise covariance using adaptive Kalman filtering (AKF) can suppress the problem of filter divergence. In Reference [26], an AKF algorithm based on maximum likelihood estimation is proposed to estimate the process noise covariance, but it is too computationally intensive. Reference [27] mentioned that the Sage–Husa adaptive filtering algorithm is simple in calculation and high in real-time, and has a good application prospect. It corrects the predicted value in the filtering process by using the observed value, and estimates the unknown observation noise. At the same time, the shortcomings of Sage–Husa filtering are obvious. It is difficult to ensure that the noise covariance matrix is positive definite in the filtering process, and the filter estimation accuracy is poor or even divergent [28]. Moreover, the convergence speed of adaptive filtering based on Sage–Husa needs to be studied. This paper proposes a coarse-to-fine tracking loop based on the weighted AKF (WAKF) algorithm. In order to reduce the WAKF initialization error and improve the convergence speed, the frequency results of acquisition are pulled in the coarse tracking stage to reduce the frequency error. After the bit synchronization is achieved, the loop switches from the coarse tracking stage to the fine tracking stage. In the fine tracking process, a tracking loop based on WAKF is used. The proposed method constructs the noise covariance matrix of the Sage–Husa algorithm with weighting factors and combines UD decomposition to ensure the positive definiteness of the noise covariance matrix in the filtering process and effectively suppress the filtering divergence, to improve the tracking accuracy of the receiver under a weak signal.

In the remainder of this paper, Section 2 presents the signal model. Section 3 presents the proposed coarse-to-fine WAKF tracking loop. In Section 4, simulations and experiments are carried out to compare different algorithms to verify the effectiveness of the proposed algorithm. Finally, Section 5 presents the conclusions.

## 2. System Model

### 2.1. Signal Model

The positioning part in CPIS adopts the DSSS-CDMA system described above, and the navigation message adopts BPSK modulation on the spreading code. The structure of the CPIS base station is shown in Figure 1. The integrated signal broadcast is realized by adding a positioning signal generator to the communication base station.

For a communication terminal, the signal it receives is a communication signal superimposed with a low-power positioning signal, and the positioning part is regarded as interference to the communication part. We define the ratio of communication signal power to positioning signal power as CPR. Based on a large number of experimental test results, we found that when CPR is greater than 18 dB, the communication QoS is not affected [18]. For the CPIS positioning receiver, the communication signal can be regarded as white Gaussian noise to the positioning signal [20]. The CPIS received by the receiver can be expressed as:(1)$$r^{\left(i\right)}\left(t\right)=\sum_{i=1}^{N_{BS}}A^{\left(i\right)}d^{\left(i\right)}\left(t\right)c^{\left(i\right)}\left(t-\tau_{i}\right)cos\left(2\pi \left(f_{c}+f_{d,i}\right)t+\varphi_{0,i}\left(t\right)\right)+\omega \left(t\right)$$
where $N_{BS}$ represents the number of base stations, $A^{\left(i\right)}$ is the signal amplitude, $d^{\left(i\right)}$ is the navigation message modulated on the positioning part, $c^{\left(i\right)}$ represents the spread spectrum sequence, $\tau_{i}$ represents the signal delay, $f_{c}$ represents the signal carrier frequency, $f_{d,i}$ represents the Doppler frequency shift, $\varphi_{0,i}\left(t\right)$ represents the initial carrier phase, and ω(t) represents the zero mean variance as $\sigma_{n}^{2}$ plus white Gaussian noise.

The CPIS is down-converted to a digital IF signal through the RF front-end:(2)$$r_{IF}{}^{\left(i\right)}\left(nT_{s}\right)=A_{IF}d^{\left(i\right)}\left(nT_{s}\right)c^{\left(i\right)}\left(nT_{s}-\tau_{i}\right)e^{j2\pi \left(f_{IF}+f_{d,i}\right)nT_{s}+\varphi_{0,i}}+\omega \left(n\right)$$
where $T_{s}$ is the sampling period, $A_{IF}$ is the IF signal amplitude, and $f_{IF}$ is the *IF* frequency.

The receiver baseband part acquires, tracks, and demodulates the *IF* signal. After acquiring the rough carrier frequency and code phase, the tracking loop initializes local parameters according to the acquisition results, and continuously tracks the signal to achieve higher-precision parameter estimation. In each tracking channel, the *IF* signal is divided into two paths, which are multiplied by the in-phase and quadrature local carriers to strip the carrier. The result is then multiplied by the local pseudocode, and the I and Q signals are generated after integration and removal. The I/Q integral values are expressed as:
(3)$$I\left(m\right)=Ad\left(m\right)sinc\left(∆fT_{coh}\right)e^{(j\left(2\pi ∆fT_{coh}+∆\varphi \right)}$$
(4)$$Q\left(m\right)=Ad\left(m\right)sinc\left(∆fT_{coh}\right)e^{(j\left(2\pi ∆fT_{coh}+∆\varphi \right)}$$
where I(m) and Q(m) are the results of the coherent integration of the mth segment, $T_{coh}$ is the coherent integration time, and A, ∆f, and ∆φ represent the coherent integration amplitude, signal residual carrier frequency, and residual carrier phase, respectively.

### 2.2. Traditional Coarse-to-Fine Carrier Tracking Scheme

The acquisition process has achieved rough estimation of the carrier frequency, and the frequency acquisition error of the CPIS positioning receiver is within 200 Hz. Due to the frequency error of acquisition, the convergence time of the tracking loop will be long or the loop will lose lock, which will affect the output of the receiver’s positioning result. In order to speed up the convergence speed, the tracking loop will be designed as a two-stage structure. The traditional coarse-to-fine tracking loop structure is shown in Figure 2.

As shown in Figure 2, the coarse tracking process includes two parts: frequency pulling and phase-locked loop. First, the receiver quickly pulls the carrier frequency based on the fast Fourier transformation (FFT) discriminator to reduce the carrier frequency error obtained by capturing. The block diagram of the FFT discriminator is shown in Figure 3. The input I/Q two-way signals are coherently accumulated to obtain the integral value and then sent to the frequency discriminator to obtain the estimated value of the residual carrier frequency. In order to eliminate the influence of the navigation message on the integration results, the coherent integration results are squared and then FFT operations are performed to obtain the power spectral density [29]. Since the formula is a single-tone signal, and the power is concentrated at a single frequency point, we can obtain an estimate of the residual carrier frequency by detecting the maximum amplitude in the frequency domain.

The integral value of I/Q in Equations (3) and (4) is affected by $∆fT_{coh}$. When the tracking loop is tracking stably, the frequency error ∆f tends to zero, so the sinc function is approximately 1, and the complex signal r(m) is rewritten as:(5)$$r\left(m\right)=Ad\left(m\right)\exp\left(j2\pi ∆f\left(t+\frac{T}{2}\right)+\delta \varphi \right).$$

The coherent integral value is squared as follows:(6)$$S\left(m\right)=r{\left(m\right)}^{2}=A^{2}\exp\left(j2\left(2\pi ∆f\left(t+\frac{T}{2}\right)+\delta \varphi \right)\right).$$

We select the coherent integral value sequence $S^{\prime }\left(m\right)=\left\{S_{1}\ldots S_{n_{FFT}}\right\}$ composed of $n_{FFT}$ points for FFT operation, and obtain the following results:(7)$$X\left(k\right)=\sum_{i=1}^{n_{FFT}}S^{\prime \left(i\right)}\exp\frac{-2\pi j\left(i-1\right)\left(k-1\right)}{n_{FFT}}$$
where X(k) is the result of frequency domain transformation, $n_{FFT}$ is the number of points processed by FFT, and $S_{p}\left(i\right)$ is the time domain complex signal.

The residual frequency estimate for the carrier is expressed as:(8)$$∆\hat{f}=\frac{1}{2}\left(k_{max}-1-\frac{n_{FFT}}{2}\right)f_{r}$$
where $k_{max}$ is the index corresponding to the FFT calculation and the maximum magnitude. The precision of the discriminant frequency of the FFT is determined by $f_{r}=1/T_{t}$, where $T_{t}$ is the time length of the coherent integral value sequence. To improve the resolution of the frequency estimation, we correct the frequency estimation error according to the following equation:(9)$$∆\hat{f}=\frac{1}{2}\left(k_{max}-1-\frac{n_{FFT}}{2}+\frac{A_{2}}{A_{1}+A_{2}}f_{r}\right).$$

The FFT discriminator outputs the residual carrier frequency ${\hat{f}}_{e}$, and initializes the second-order PLL. At this time, the estimated error of the residual carrier frequency is relatively small, and the PLL can quickly converge and stably track the residual carrier frequency. The PLL tracking loop has the characteristics of constant parameters. The design of the loop parameters of the PLL needs to balance between reducing noise and dynamic response, so the tracking accuracy of the PLL is limited. Moreover, when the quality of the occluded signal of the receiver deteriorates, the error of the result output by the PLL becomes larger or even loses the lock. To cope with this problem, a KF is usually selected in the fine tracking stage to improve the robustness and tracking accuracy of carrier tracking. After achieving bit synchronization, the receiver can switch to the KF-based fine tracking stage. The carrier tracking loop based on the Kalman filter can further reduce the error of carrier tracking and improve the robustness under weak signal reception. Based on the KF tracking algorithm, the state vector can be modeled as:(10)$$X_{k}={\left[\begin{array}{c} x_{∆\varphi }\;\;\;\;x_{∆\omega }\;\;\;\;x_{∆a} \end{array}\right]}^{T}$$
where $x_{∆\varphi }$ is the phase difference between the actual carrier and the local carrier, $x_{∆\omega }$ is the difference between the actual carrier and the local carrier frequency, and $x_{∆a}$ is the rate of change of the carrier frequency.

The state update equation is expressed as follows:
(11)$$X_{k+1}=\Phi X_{k}+w_{k}=\left[\begin{array}{ccc} 1 & T & \frac{T^{2}}{2}\\ 0 & 1 & T\\ 0 & 0 & 1 \end{array}\right]{\left[\begin{array}{c} x_{∆\varphi }\\ x_{∆\omega }\\ x_{∆a} \end{array}\right]}_{k}+{\left[\begin{array}{c} w_{1}\\ w_{2}\\ w_{3} \end{array}\right]}_{k}$$
where $w_{k}$ is the state noise error, the covariance is Q, and T is the period of the tracking loop update.

The observation equation is written as:(12)$$z_{k}=H_{k}X_{k}+n_{k}=\left[1,\frac{T}{2},\frac{T^{2}}{6}\right]{\left[\begin{array}{c} x_{∆\varphi }\\ x_{∆\omega }\\ x_{∆a} \end{array}\right]}_{k}+n_{k}$$
where $z_{k}$ is the observation value of the phase detector output, $H_{k}$ is the observation matrix, the observation noise is $n_{k}$, and the covariance is $R_{k}$.

## 3. Proposed Weighted Adaptive Kalman-Filter-Based Coarse-To-Fine Carrier Tracking Loop

As mentioned above, the traditional coarse-to-fine tracking loop tracking completes the pulling of the carrier frequency in the coarse tracking stage, which significantly reduces the frequency estimation error, which shortens the convergence speed of the KF-based fine tracking process. However, the KF tracking loop relies on accurate prior noise covariance, and when the environment deteriorates, the estimation accuracy of the KF becomes poor and even diverges. In order to improve the tracking accuracy of the CPIS receiver under a weak signal, this chapter proposes a carrier tracking algorithm with weighted adaptive filtering from coarse to fine to improve the performance of the receiver under a weak signal.

### 3.1. WAKF-Based Fine Carrier Tracking Algorithm

Sage–Husa adaptive filtering is an adaptive filtering algorithm proposed on the basis of a KF. The algorithm solves the problem of filter divergence caused by fixed covariance by estimating the noise covariance in real time in the recursive process. The Sage–Husa adaptive filtering process is described as follows [28]:(13)$${\hat{X}}_{k,k-1}=\Phi {\hat{X}}_{k-1}+{\hat{q}}_{k-1}$$
(14)$$P_{k,k-1}=\Phi P_{k-1}\Phi^{T}+{\hat{Q}}_{k}$$
(15)$$K_{k}=P_{k,k-1}H_{k}^{T}{\left(H_{k}P_{k,k-1}H_{k}^{T}+{\hat{R}}_{k}\right)}^{-1}$$
(16)$${\hat{X}}_{k}={\hat{X}}_{k,k-1}+K_{k}\left(z_{k}-H_{k}{\hat{X}}_{k,k-1}-{\hat{r}}_{k}\right)$$
(17)$$P_{k}=\left(I-K_{k}H_{k}\right)P_{k,k-1}$$
where ${\hat{X}}_{k}$ is the estimated value of $X_{k}$, ${\hat{X}}_{k,k-1}$ is the predicted value of $X_{k}$, $P_{k}$ is the estimated error covariance, $P_{k,k-1}$ is the predicted error covariance, ${\hat{q}}_{k-1}$ is the estimated mean of $w_{k}$, and ${\hat{r}}_{k}$ is the estimated mean of $n_{k}$. ${\hat{q}}_{k}$, $\;{\hat{Q}}_{k}$, $\;{\hat{r}}_{k}$, and ${\hat{R}}_{k}$ are obtained recursively according to the following equations:(18)$${\tilde{z}}_{k}=z_{k}-H_{k}{\hat{X}}_{k,k-1}$$
(19)$${\hat{q}}_{k}=\left(1-d_{k}\right){\hat{q}}_{k-1}+d_{k}\left({\hat{X}}_{k}-\Phi {\hat{X}}_{k-1}\right)$$
(20)$${\hat{Q}}_{k}=\left(1-d_{k}\right){\hat{Q}}_{k-1}+d_{k}\left(K_{k}{\tilde{z}}_{k}{\tilde{z}}_{k}^{T}K_{k}^{T}+P_{k}-\Phi P_{k-1}\Phi^{T}\right)$$
(21)$${\hat{r}}_{k}=\left(1-d_{k}\right){\hat{r}}_{k-1}+d_{k}\left(z_{k}-H_{k}{\hat{X}}_{k,k-1}\right)$$
(22)$${\hat{R}}_{k}=\left(1-d_{k}\right){\hat{R}}_{k-1}+d_{k}\left({\tilde{z}}_{k}{\tilde{z}}_{k}^{T}-H_{k}P_{k,k-1}H_{k}^{T}\right)$$
where $d_{k}=\left(1-b\right)/\left(1-b^{k+1}\right)$, *b* is the forgetting factor used to control the weight of the observation value, and its value range is usually between 0.95 and 0.99.

It can be seen from Equations (20) and (22) that the noise covariance is difficult to maintain as positive definite in the iterative process. And in the algorithm update process, the filter gain $K_{k}$ will be affected when the inverse of the matrix $H_{k}P_{k,k-1}H_{k}^{T}+{\hat{R}}_{k}$ does not exist or the error is large. This will eventually cause ${\hat{X}}_{k}$ divergence.

To ensure that the Q matrix is positive definite, the iteration equation of Q is modified as follows [27]:(23)$${\hat{Q}}_{k}=\left(1-d_{k}\right){\hat{Q}}_{k-1}+d_{k}\left(K_{k}{\tilde{z}}_{k}{\tilde{z}}_{k}^{T}K_{k}^{T}\right)$$

In the application process, the following filter divergence judgment criteria are designed according to ${\tilde{z}}_{k}$:(24)$$E\left[{\tilde{z}}_{k}{\tilde{z}}_{k}^{T}\right]=H_{k}P_{k,k-1}H_{k}^{T}+{\hat{R}}_{k}$$
(25)$${\tilde{z}}_{k}{\tilde{z}}_{k}^{T}\geq \eta \mathrm{tr}\left(E\left[{\tilde{z}}_{k}{\tilde{z}}_{k}^{T}\right]\right)$$

η is a variable coefficient, η ≥ 1. The convergence condition is the strictest when η=1. When both sides of the equation are equal, the filter is judged to be converged. When the equation is true, the observation noise covariance ${\hat{R}}_{k}$ needs to be adaptively adjusted to reduce the estimation error of the filter. We propose the following weighting factors $f_{k}\left(v_{k}\right)$ to adaptively adjust ${\hat{R}}_{k}$:(26)$$f_{k}\left(v_{k}\right)=\alpha^{v_{k}-1}$$
(27)$$v_{k}=\frac{{\tilde{z}}_{k}{\tilde{z}}_{k}^{T}}{\mathrm{tr}\left(E\left[{\tilde{z}}_{k}{\tilde{z}}_{k}^{T}\right]\right)}$$
where 1<α<2. Therefore, the ${\hat{R}}_{k}$ update equation can be rewritten as:(28)$${\hat{R}}_{k}=f_{k}\left(v_{k}\right){\hat{R}}_{k-1}$$

It can be seen from Equation (28) that when $v_{k}$ is greater than 1, $f_{k}\left(v_{k}\right)$ is greater than 1, and ${\hat{R}}_{k}$ increases. $v_{k}$ is less than 1, $f_{k}\left(v_{k}\right)$ is less than 1, and ${\hat{R}}_{k}$ decreases. $R_{0}$ is positive definite and the weighting factor $f_{k}\left(v_{k}\right)$ is greater than 0, so the observation noise covariance ${\hat{R}}_{k}$ remains positive definite during the update.

Due to the limitation of the expression precision of numbers in practical applications, round-off errors will occur, and the covariance matrix cannot guarantee positive definiteness. We introduce the UD decomposition algorithm to decompose the error covariance moment matrix *P* into the upper triangular matrix and the diagonal matrix *D* during the filter update process [30]. Due to the special structure of the matrix *U* and the matrix *D*, the errors will not accumulate and avoid the non-positive definiteness of the noise covariance. In the filter update period, the error covariance matrix $P_{k,k-1}$ is first decomposed by UD.
(29)$$P_{k,k-1}=UDU^{T}$$

Substituting Equation (29) into Equation (24), we obtain $\;E\left[{\tilde{z}}_{k}{\tilde{z}}_{k}^{T}\right]$:(30)$$E\left[{\tilde{z}}_{k}{\tilde{z}}_{k}^{T}\right]=H_{k}UDU^{T}H_{k}^{T}+{\hat{R}}_{k}$$

Then, we calculate the filter gain $K_{k}$:(31)$$K_{k}=UDU^{T}H_{k}^{T}E{\left[{\tilde{z}}_{k}{\tilde{z}}_{k}^{T}\right]}^{-1}.$$

Finally, the mean square error matrix $P_{k}$ is calculated according to the filter gain $K_{k}$ and the error covariance matrix $P_{k,k-1}$.
(32)$$P_{k}=UDU^{T}-UDU^{T}H_{k}^{T}E{\left[{\tilde{z}}_{k}{\tilde{z}}_{k}^{T}\right]}^{-1}UDU^{T}.$$

This paper combines the weighting factor adaptive adjustment covariance matrix and the UD decomposition to ensure the positive definiteness of the noise co-equations Q and R in the filtering update process.

### 3.2. Coarse-to-Fine WAKF-Based Carrier Tracking Algorithm

Figure 4 shows the coarse-to-fine WAKF-based carrier loop structure. The I and Q integral results first enter the coarse tracking loop, which reduces the tracking accuracy of the frequency to achieve bit synchronization and improve the positioning speed. The carrier frequency error acquired by the communication-positioning integrated signal receiver is relatively large, and it is necessary to first reduce the frequency to the tracking range of the phase-locked loop. Figure 3 shows that based on the FFT frequency pulling algorithm, FFT frequency discrimination can quickly and accurately obtain the residual carrier frequency, and reduce the convergence time of the phase-locked loop to achieve bit synchronization. When the bits are synchronized, a switch is made to the fine tracking loop based on the AKF algorithm to achieve more accurate tracking of the carrier. The implementation steps of the whole algorithm are as follows:Initialize FFT points $n_{FFT}$ in the coarse tracking stage, and the bandwidth $B_{PLL}$ of the second-order phase-locked loop;The first step of coarse tracking: First, The times of loop updates in the carrier frequency pulling process is configured as $n_{FFT}$, and the loop stores the integral values of the I/Q channels in sequence every $T_{Coh,1}$, and obtains the integral value sequence $S^{\prime }\left(m\right)=\left\{S_{1}\ldots S_{n_{FFT}}\right\}$ of length $n_{FFT}$ and then performs an FFT operation on the sequence S′(m) to obtain the index corresponding to the maximum amplitude, and obtain the residual carrier frequency estimation value according to Formula (9). The second step of rough tracking: after completing the frequency pulling, the loop enters the second-order phase-locked loop for stable tracking, and the loop adjusts the carrier NCO every $T_{Coh,1}$;The coarse tracking stage ends when the bit synchronization is successful. The tracking loop can enter the fine tracking loop based on the WAKF algorithm from the coarse tracking stage. If bit synchronization is unsuccessful, the loop remains in the coarse tracking stage.The loop enters the fine tracking stage. The process noise covariance matrix $Q_{0}$ and the measurement noise covariance matrix $R_{0}$, the state vector matrix $X_{0}$, and the state vector error covariance matrix $P_{0}$ are initialized. The loop based on the WAKF algorithm updates the loop every integration time $T_{Coh,2}$. After the bit synchronization is successful, the integration value is not affected by the navigation data, and the integration time can be lengthened to realize the tracking of weak signals. The estimation of carrier phase and carrier frequency is achieved by the previously mentioned WAKF algorithm iterative equations. The carrier NCO performs loop adjustment according to the result output by the WAKF algorithm.

## 4. Simulation and Analysis

The coarse-to-fine WAKF-based tracking algorithm was primarily developed to improve carrier tracking performance in a weak environment. In this section, simulation and real data tests are performed to verify the reliability of the proposed algorithm. Moreover, the results of the proposed algorithm are compared to algorithms such as the coarse-to-fine AKF-based tracking algorithm, coarse-to-fine KF-based tracking algorithm, and KF-based tracking loop without a coarse tracking stage.

### 4.1. Simulation and Real Data Tests

#### 4.1.1. Tracking Sensitivity

In order to evaluate the tracking sensitivity, we simulated the three tracking algorithms (coarse-to-fine KF-based tracking algorithm, coarse-to-fine AKF-based tracking algorithm, coarse-to-fine WAKF-based tracking algorithm) and compared the three algorithms in tracking accuracy at different signal-to-noise ratios (SNRs). The simulation parameters of the algorithm were set as shown in Table 1. The SNR of the simulated signal input to the tracking loop is shown in the orange curve in Figure 5. The simulated signal SNR started from −25 dB, and the SNR ratio decreased by 1 dB every 10 s until the SNR dropped to −45 dB, and the simulation time was 210 s in total. Figure 5 also shows the estimation accuracy of the participating carrier frequencies for the three algorithms. As the signal becomes weaker, the three algorithms show the tracking accuracy of different carrier frequencies. As shown in the red curve in Figure 5, the coarse-to-fine KF-based tracking algorithm first loses the lock on the carrier frequency, and the SNR is about −35 dB at this time. The cyan curve is the tracking result of the coarse-to-fine AKF-based tracking algorithm, which shows better performance than the coarse-to-fine KF-based algorithm. The black curve shows that the coarse-to-fine WAKF-based tracking algorithm proposed in this paper loses lock when the SNR is −41 dB, so it can be seen from Figure 5 that the sensitivity of the tracking algorithm proposed in this paper is significantly better than the previous two algorithms.

In order to further verify the effectiveness and tracking performance of the proposed algorithm, Monte Carlo simulations were carried out, each SNR simulation was performed 1000 times, the residual carrier frequency was set to the same value each time, and the tracking accuracy of the above three algorithms under different signal-to-noise ratios was analyzed. The carrier frequency RMSE shown in Figure 6 was obtained. It can be seen from the results in the figure that with the decrease in the signal-to-noise ratio, the tracking accuracy of the three algorithms has decreased. As shown by the blue curve, the coarse-to-fine WAKF-based algorithm proposed in this paper has the highest accuracy under weak signals. The yellow curve indicates that the coarse-to-fine KF-based algorithm has the largest error under weak signals. The coarse-to-fine WAKF-based algorithm proposed in this paper improves the tracking accuracy of the tracking loop under weak signals.

Theprobability of successful tracking of simulations is defined as tracking probability in the literature [31]. The tracking result satisfies that the 1σ carrier estimated frequency error is less than 5 Hz and the maximum frequency error is less than 20 Hz within 10 s as successful tracking, and the probability of successful tracking is considered. The SNR corresponding to more than 50% is defined as tracking sensitivity. Figure 7 shows the tracking sensitivity of the three algorithms; the tracking sensitivity of the coarse-to-fine KF-based algorithm is −37 dB, followed by the coarse-to-fine AKF-based tracking algorithm at −39 dB, which is 2 dB higher than the KF-based tracking algorithm. The coarse-to-fine WAKF tracking sensitivity is −41 dB, which is 2 dB higher than AKF-based tracking algorithm. From the simulation results, the proposed coarse-to-fine WAKF-based tracking algorithm significantly improves the tracking sensitivity of the communication-positioning integrated signal receiver under weak signals.

#### 4.1.2. Convergence Speed

In order to analyze the convergence speed, we compare the proposed coarse-to-fine WAKF tracking algorithm and the WAKFtracking algorithm without coarse tracking stage, which are denoted as Algo1 and Algo2 in Figure 8, respectively. The difference between the two algorithms is whether there is a coarse tracking stage. The signal-to-noise ratio is set to −25 dB and −35 dB, and the parameter settings of the coarse-to-fine WAKF based tracking algorithm refer to Table 1. The parameter configuration of the WAKF based tracking algorithm is the same as that of the fine tracking stage of the proposed algorithm. From the carrier phase tracking results in Figure 8 and Figure 9, it can be seen that both algorithms can converge in the end, while the Algo2 tracking algorithm has a slower convergence rate because the initial value of Algo2 is from the acquisition. It can be seen from the figure that the convergence speed of the Algo1 algorithm is about 1 s faster than that of Algo2 algorithm. When the signal-to-noise ratio in Figure 9 is set to −35 dB, the convergence speed of the Algo2 algorithm is improved by about 1.5 s. The coarse-to-fine WAKF-based tracking algorithm proposed in this paper reduces the large frequency error caused by acquisition in the coarse tracking stage, and improves the convergence speed of the tracking loop.

### 4.2. Real Data Tests

We conducted tests in a real environment to further verify the effectiveness of the new algorithm. We built the communication-positioning integrated signal receiver sensitivity test environment as shown in the Figure 10. The communication and navigation fusion base station is mainly composed of a communication base station, a positioning base station, and a clock distributor. The communication base station is responsible for generating 5G communication signals, the positioning base station generates the above-mentioned positioning signals, and the communication and navigation fusion signals work in the 3.5 GHz frequency band. The clock distributor provides the reference clock signal and the second pulse for the communication base station and the positioning base station. After the fusion of the two signals is controlled by the stepping attenuator, it enters the positioning receiver for baseband signal processing. The positioning receiver developed by us is used in the test. As shown in the Figure 11, the observation value of the loop output is transmitted to the cell phone. The following Table 2 shows the tracking sensitivity test results for the three algorithms. It can be seen from the table that the algorithm proposed in this paper can track the sensitivity by 2–4 dB compared with the other two, and can track the communication-positioning integrated signal with a signal strength of −137 dBm.

In addition, we conducted a test in a shopping mall in Tianjin, and the test scenario shown in Figure 12a is a floor plan of the mall. The blue point is the location of the communication-positioning integrated base station, and the red point represents the test point. Figure 12b is a photograph of the test scenario. The signal is weak due to many indoor obstructions, which puts forward requirements for the high sensitivity of the receiver. We selected 20 test points on the first floor of the mall, used the total station to establish a self-built coordinate, and mapped the accurate coordinates of all test points. We placed the positioning terminal at each test point for 5 min, compared the positioning results with the coordinates calibrated by the total station, and counted 1σ positioning accuracy. Figure 13 shows the CDFs of two test points. Test point 1 and test point 2 represent the positioning statistical results in the case of line of sight (LOS) and non-line of sight (NLOS), respectively. It can be seen from the results that the positioning accuracy of LOS of the algorithm proposed in this paper is the highest compared with the other two algorithms. In the case of NLOS, the WAKF-based algorithm still maintains good positioning accuracy, and the accuracy of the other two algorithms decreases significantly. The above results verify the effectiveness of the proposed algorithm in an actual environment.

## 5. Conclusions

In order to improve the convergence speed and tracking accuracy of the tracking loop of a weak communication navigation fusion signal, a coarse-to-fine weighted adaptive-filter-based carrier loop algorithm was proposed in this paper. The algorithm combines the weighting factor and UD decomposition to ensure the positive definiteness of the covariance matrix of the Sage–Husa filter update process and suppress the filter divergence. Compared with the carrier loop based on a KF and the Sage–Husa adaptive filtering loop, it improves the tracking accuracy of the weak signal carrier. Moreover, the coarse tracking stage significantly reduces the initial value error of the weighted adaptive filtering algorithm and improves the filtering convergence speed. Simulation and practical tests have verified the effectiveness of the proposed algorithm. From the results, compared with current algorithms, the coarse-to-fine tracking structure shortens the filtering convergence speed. The divergence of the adaptive filter loop is effectively suppressed, and the tracking sensitivity and positioning accuracy are improved.

## Figures and Tables

**Figure 1 sensors-22-04068-f001:**
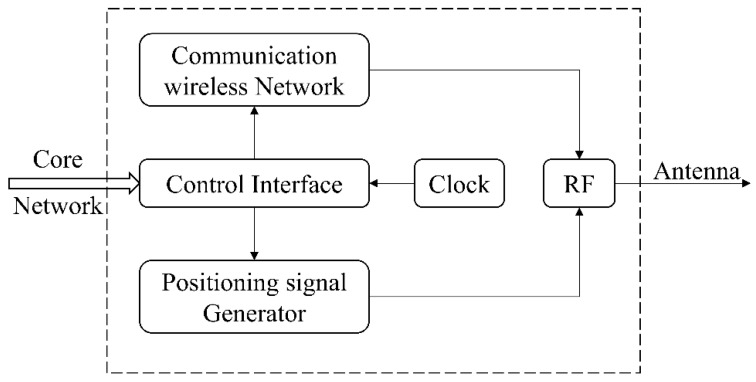
Structure of the CPIS base station.

**Figure 2 sensors-22-04068-f002:**
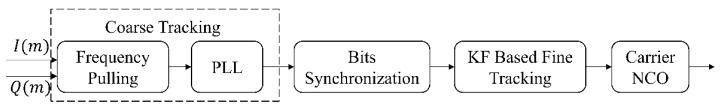
Traditional coarse-to-fine tracking loop structure.

**Figure 3 sensors-22-04068-f003:**
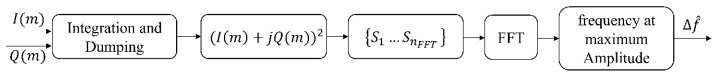
FFT discriminator block diagram.

**Figure 4 sensors-22-04068-f004:**
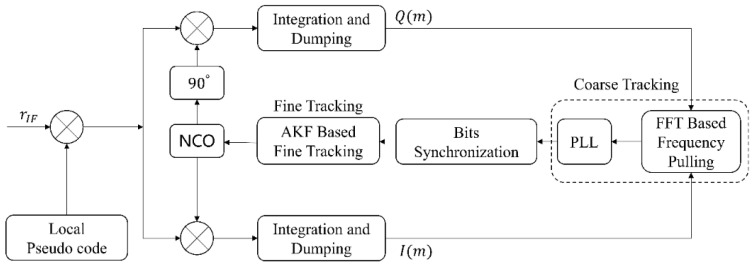
Structure of the proposed coarse-to-fine WAKF-based carrier loop.

**Figure 5 sensors-22-04068-f005:**
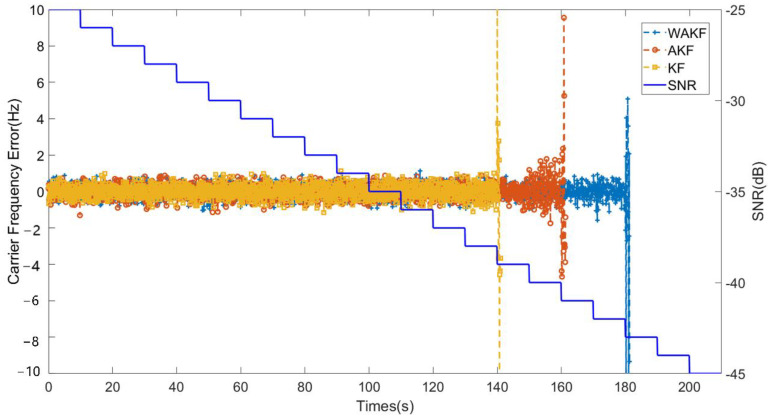
Tracking results under weak signal.

**Figure 6 sensors-22-04068-f006:**
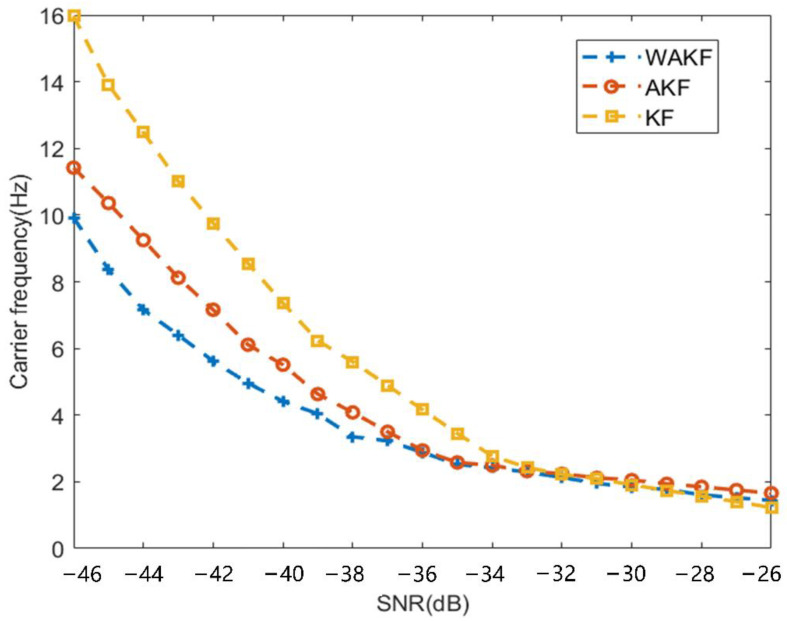
Frequency estimation errors of three algorithms under different SNRs.

**Figure 7 sensors-22-04068-f007:**
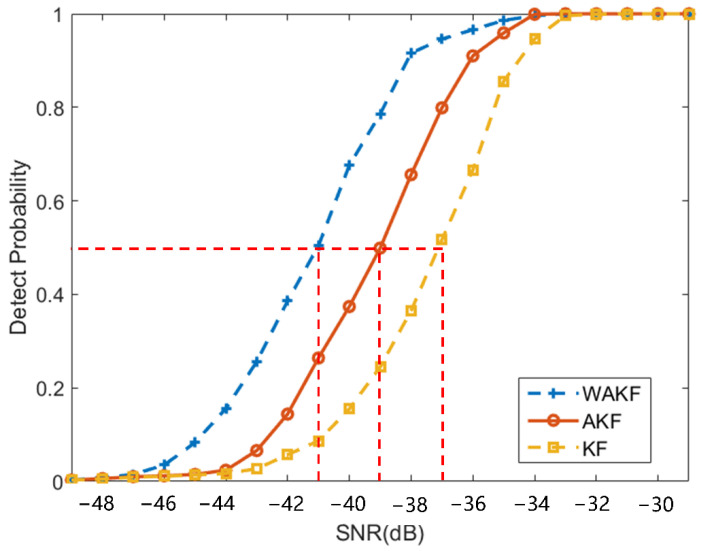
Comparison of tracking probability of three algorithms.

**Figure 8 sensors-22-04068-f008:**
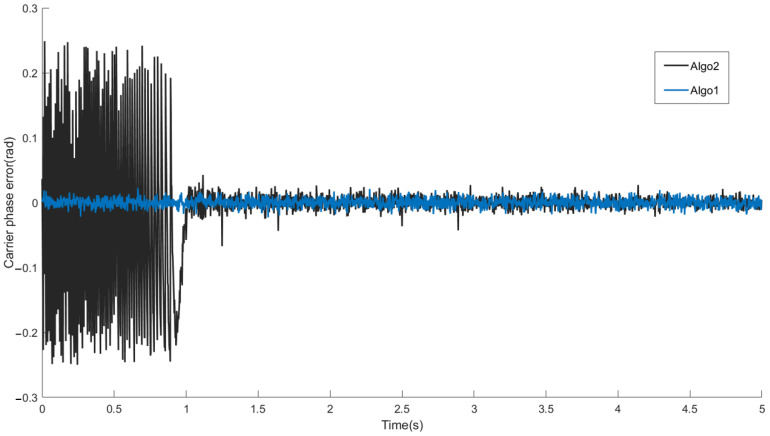
Carrier phase tracking result, SNR = −25 dB.

**Figure 9 sensors-22-04068-f009:**
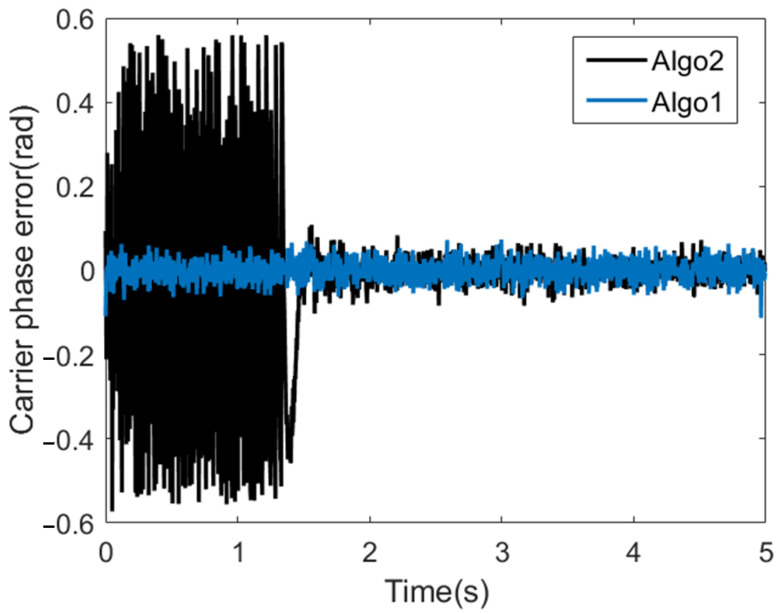
Carrier phase tracking result, SNR = −35 dB.

**Figure 10 sensors-22-04068-f010:**
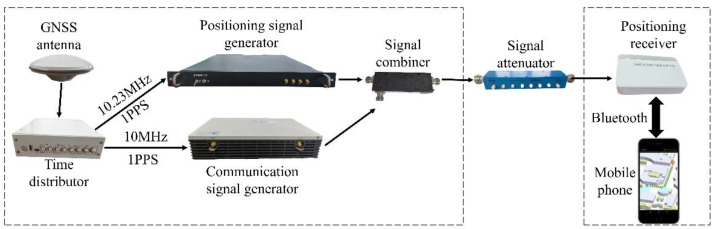
Test environment.

**Figure 11 sensors-22-04068-f011:**
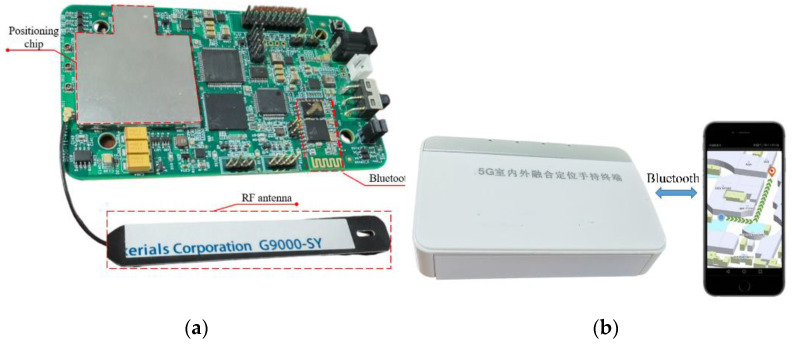
CPIS receiver. (**a**) is the internal structure of the CPIS receiver and (**b**) is the data transmission protocol between receiver and mobile phone.

**Figure 12 sensors-22-04068-f012:**
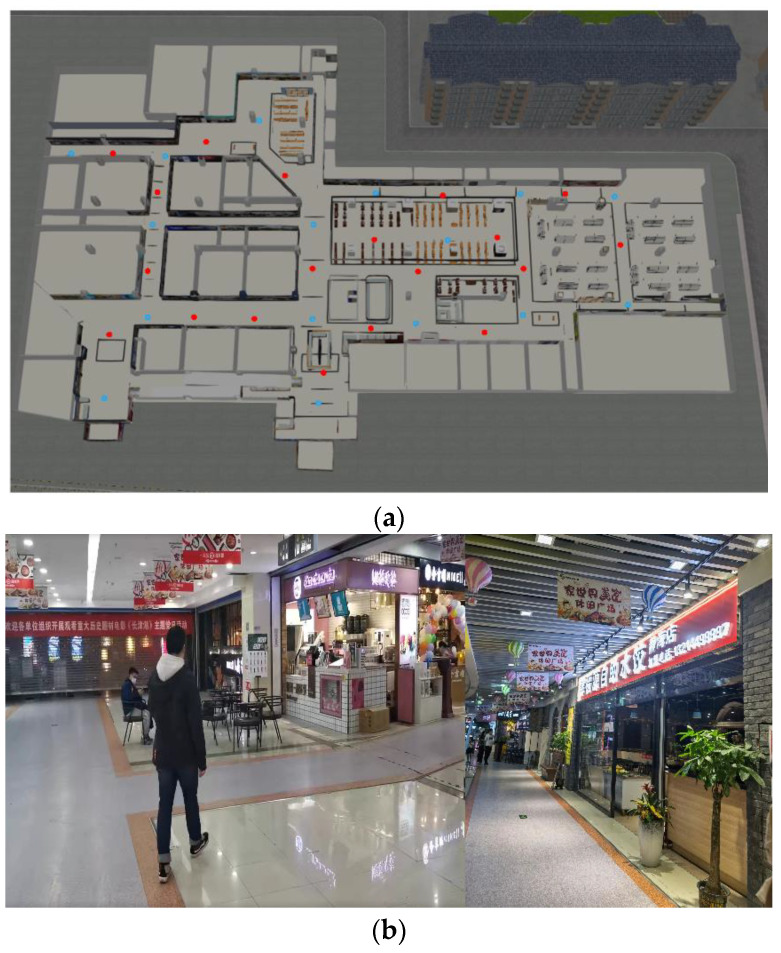
Test scenario. (**a**) is floor plan of the mall. The blue point is the location of the communication-positioning integrated base station, and the red point represents the test point. (**b**) is a photograph of the test scenario.

**Figure 13 sensors-22-04068-f013:**
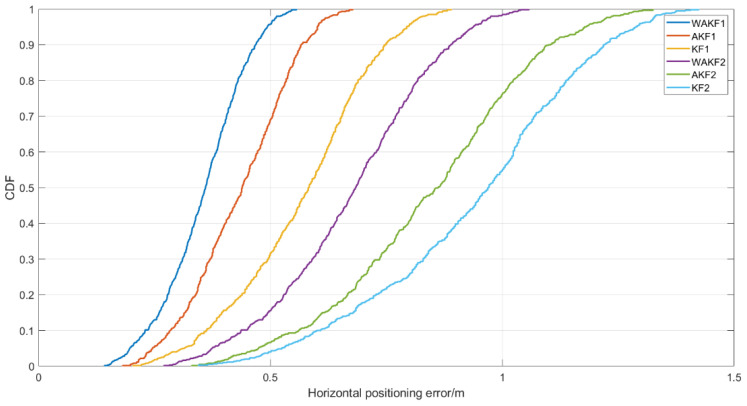
The positioning accuracy error in the horizontal direction.

**Table 1 sensors-22-04068-t001:** Simulation parameters.

Parameter	Value
Length of PRN code	10,230
Code rate	10.23 MHz
Sampling rate	50 MHz
Noise baseband of PLL in coarse tracking stage	10 Hz
$T_{Coh,1}$ in coarse tracking stage	5 ms
$T_{Coh,2}$ in fine tracking stage	10 ms

**Table 2 sensors-22-04068-t002:** Tracking sensitivity of three algorithms under different SNRs.

Signal (dBm)	−128	−129	−130	−131	−132	−133	−134	−135	−136	−137
WAKF	Y ^1^	Y	Y	Y	Y	Y	Y	Y	Y	Y
AKF	Y	Y	Y	Y	Y	Y	Y	Y	- ^2^	-
KF	Y	Y	Y	Y	Y	Y	-	-	-	-

^1^ The mark Y represents tracking under corresponding signal power. ^2^ N/A means losing tracking under corresponding signal power.

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
