# Peer review of "A Novel Carrier Loop Based on Coarse-to-Fine Weighted Adaptive Kalman Filter for Weak Communication-Positioning Integrated Signal"

_sensors, 2022, doi:10.3390/s22114068_

Round 1

Reviewer 1 Report

Thank you for sharing this good work.

I only have a few remarks:

  • Fig. 12: blue and red dots are not intuitive. Please add a legend or subtitle such that the reader can more easily (without reading the full text) get an idea how the experiment was conducted.
  • Photograph(s) of the test scenario would be great to get an impression of the complexity of the scenario. SNR/receive signal strength are certainly not the only factors affecting performance.
  • More CDF's of different and especially critical test spots would be interesting. Most likely the probalitiy densities will vary depending if the test spot was at a very "good" or "bad" location, so the "CDF spread" between the "best" and "worst" case would be interesting here. (another indication of method robustness)
  • just some typos: "SRN" vs SNR, "PCR" vs CPR, "Detect Probality", "losting"

Reviewer 2 Report

This paper proposes a coarse-to-fine tracking loop based on the weighted AKF algorithm. Also this paper combines the weighting factor adaptive adjustment covariance matrix and the UD decomposition to ensure the positive definiteness of the noise co-equations Q and R in the filtering update process

It is very well written as it analyzes successfully all the parameters of indoor positioning.

I believe that the authors need to pay more attention to their conclusions. They need to further analyze the results of their research in the conclusions.

After this correction, I believe that the paper is ready for publication

In my opinion, in the paper symbols and abbreviations should be included

Reviewer 3 Report

The paper provides an interesting and valuable contribution to the literature.  I recommend its publication.

However, there are several changes to made in the English presentation, which can be readily accomplished.

  1. In both the title and the first line of the Abstract, the word "Intergrated" appears.  It should be "Integrated".
  2. On p. 2 (lines 56, 59, 60):  After the abbreviation i.e., the words should not be capitalized.  
  3. Throughout the paper, there are many instances where an equation is presented, followed by a paragraph indentation and the beginning of a new sentence with the word "Where."  This is incorrect in several ways.  First, the equation should be viewed as a clause within the sentence, and the equation should be punctuated with a comma.  Second, the sentence continues with the word "where", which should NOT be capitalized.  As stated (incorrectly)in the paper, "Where..." begins a dependent clause, and cannot form the first word of a sentence.  Third, the new paragraph indentation should not be used --- it is all part of the sentence, so a new paragraph cannot be starting.  This problem appears (at least) in p. 3 eq (1), p. 4 eq(2), p. 4  eq (4), p. 5 eq (8), p. 6  eq (11), p 6 eq 12, p 6 eq (17), p 7 eq (22), p 7 eq (25), p 7 eq (27), p 8 eqs (29).
  4. p. 3 line 128: tau_i represents the signal _delay_.
  5. p. 3 line 127: It seems you want to say "N_{BS} represents the number of base stations"
  6. p. 4 line 149: remove the word "even"
  7. p 5 line 168: Something is strange here -- a pair of sentences got mangled together.
  8. p. 6 lines 203: the word "Propesed" in the section heading is misspelled.
  9. p. 7 lines 235:  I don't understand what you mean by "All propose"
  10. p. 7 line 245:  I don't understand what you mean by "by mistake"
  11. p. 8 equation (31):  This ends a sentence, and so should be punctuated with a period.  In fact, you need to check that all of your sentences are punctuated correctly, with periods at the ends of sentences, and commas leading to clauses explaining the notation (as described above).  Also, the "Finally" in the next line should probably be in the same paragraph (no paragraph indentation).
  12. p. 9 line 305: there should be no space before a comma.
  13. p. 11 line 352: Put period, not comma, at end of sentence.
  14. p. 14, line 394.  The mall floorplan is in figure 12, not 11.
  15. p. 14 In this figure are a lot of red points and blue points.  Your description in the paragraph is confusing.
  16. p. 15 line 410:  Should read "The paper combines ..."
  17. References: IEEE should be all capitalized, in all the references (e.g, 1, 2, 3, ...)
  18.  
